# *MK3* Gene Upregulates Granulosa Cell Apoptosis Through the TNF/P38 MAPK Pathway in Chicken

**DOI:** 10.3390/cells14201630

**Published:** 2025-10-20

**Authors:** Li Chen, Jia Liu, Ying Zhang, Jinsong Pi, Yan Wu

**Affiliations:** 1Institute of Animal Husbandry and Veterinary Medicine, Hubei Academy of Agricultural Sciences, Wuhan 430064, China; 17671876856@163.com (L.C.); 17390805127@163.com (J.L.); zy513213743@126.com (Y.Z.); pijinsong@sina.com (J.P.); 2College of Animal Science and Technology, Yangtze University, Jingzhou 434025, China; 3Hubei Key Laboratory of Animal Embryo Engineering and Molecular Breeding, Wuhan 430064, China

**Keywords:** *MK3* gene, follicular development, granulosa cells, cell apoptosis

## Abstract

In poultry production, the laying rate is a critical economic trait, as high egg production significantly enhances profitability. Mitogen-activated protein kinase-activated protein kinase 3 (*MK3*) is a member of the mitogen-activated protein kinase (MAPK) family, which plays an important role in follicular development. Our previous RNA-seq analysis revealed that *MK3* expression was significantly altered in the ovaries of laying hens exposed to normal versus light-deprivation conditions. Based on previous RNA-seq analysis of chicken ovaries, this study focused on the *MK3* gene to explore its role in regulating apoptosis of follicular granulosa cells in laying hens. The results demonstrated that *MK3* overexpression induced granulosa cell apoptosis by modulating the expression of key proliferation- and apoptosis-related genes, including *FAS*, *Caspase3*, *BCL2*, and *C-myc*. These findings were further validated using specific siRNA-mediated knockdown of *MK3*. Flow cytometry, CCK-8, and EdU assays consistently showed that *MK3* facilitated apoptosis and inhibited granulosa cell proliferation. Additionally, dual-luciferase reporter assays revealed that the transcription factor *WT1* bound to the *MK3* promoter and enhanced its transcriptional activity. Mechanistically, *MK3* regulated granulosa cell apoptosis through the TNF/P38 MAPK pathway. This conclusion was corroborated by treatment with the P38 inhibitor GS-444217 and specific siRNA targeting components of the pathway. In summary, *MK3* promotes granulosa cell apoptosis in the follicles of laying hens, is transcriptionally regulated by *WT1*, and exerts its pro-apoptotic effects via the TNF/P38 MAPK pathway.

## 1. Implications

In recent years, the poultry industry has experienced rapid growth. Rising living standards and evolving consumer preferences have increased the demand for high-quality local poultry breeds with distinctive traits. However, low reproductive efficiency remains a major constraint on the sustainable development of the poultry industry. In this study, we investigated the role of the mitogen-activated protein kinase *-activated protein kinase 3* (*MK3* or *MAPKAPK3*) gene in regulating granulosa cell apoptosis in the ovarian follicles of laying hens. Our findings highlight the key role of *MK3* in follicular atresia, providing novel insights to explore the molecular mechanism of follicle development and production performance in chickens.

## 2. Introduction

The endocrine mechanisms of the hypothalamic-pituitary-gonadal (HPG) axis have become increasingly well understood, and granulosa cells have consistently been recognized as central regulators of follicular growth, development, and atresia [1]. Advances in molecular biology have significantly deepened our understanding of folliculogenesis and egg production in poultry, especially in relation to granulosa cell apoptosis, which plays a pivotal role in determining egg-laying performance. Apoptosis of granulosa cells has been identified as the primary mechanism driving follicular atresia [2]. Moreover, granulosa cells secrete various bioactive molecules, including growth factors, cytokines, and gonadal steroids, which are essential for the transition of primordial follicles into mature, ovulation-ready follicles [3]. Normal follicular development is a key determinant of egg production in poultry, and the physiological status of granulosa cells profoundly affects follicle maturation. Therefore, elucidating the mechanisms that govern granulosa cell function holds great promise for enhancing reproductive efficiency in poultry.

*MK3*, a member of the mitogen-activated protein kinase (MAPK) family, is a serine/threonine protein kinase that transduces extracellular signals into diverse intracellular responses [4]. It is broadly expressed across animal species and participates in various physiological processes, including cytokine production, endocytosis, cell cycle regulation, autophagy, inflammation, endometrial hyperplasia, chromatin remodeling, and transcriptional control [5,6]. Our previous RNA-seq analysis revealed that *MK3* expression was significantly altered in the ovaries of laying hens exposed to normal versus light-deprivation conditions. Studies in livestock have primarily associated *MK3* with the regulation of follicular atresia, follicle growth and development, corpus luteum formation and regression, and oocyte maturation [7,8,9,10]. However, most existing research on *MK3*-mediated regulation of follicular development, granulosa cell proliferation, and apoptosis has been conducted in mammalian models, such as pigs and mice. To date, investigations into the function of *MK3* in avian reproductive traits remain limited. Therefore, elucidating the role of *MK3* in the reproductive biology of laying hens is crucial.

In this study, based on the RNA-seq results of ovaries for normal- and light-deprivation conditions, we selected the different expression gene *MK3* as the target gene to clarify its role in follicular development of chicken. Our work aims to confirm the molecular mechanism by which *MK3* modulate follicular development when light changes and provide a novel insight into the regulation of production performance in chickens.

## 3. Methods

### 3.1. Experimental Animals

All animal experiments were conducted in accordance with the guidelines set forth by the National Administration of Laboratory Animal Management (Ministry of Science and Technology of China, 2017). The protocol for animal experiments was approved by the Animal Care and Use Committee of the Institute of Animal Husbandry and Veterinary Medicine, Hubei Academy of Agricultural Sciences (2024-620-000-001-015, 2024). Experimental animals were obtained from the Poultry Breeding and Farming Research Institute of the Hubei Academy of Agricultural Sciences. Follicles of various grades were collected from healthy laying hens (Jing Tint 6 laying hens).

### 3.2. Isolation, Culture, Transfection, and Dual Luciferase Reporter Gene Assay of Chicken Follicular Granulosa Cells

Follicles of various grades were collected from actively Jing Tint 6 laying hens and immediately placed in phosphate-buffered saline (PBS). After follicle dissection, capillaries and mucous membranes were removed. The follicles were washed with PBS containing a triple-antibiotic solution, punctured, and the inner membrane was inverted to facilitate mechanical separation of granulosa cells. Following centrifugation, cells were digested with trypsin for 2 min to halt enzymatic activity, then filtered through a 200-mesh strainer and washed again by centrifugation. The isolated granulosa cells were resuspended in a culture medium composed of M199 supplemented with 10% fetal bovine serum (FBS), 1% penicillin-streptomycin, and amphotericin B. Cell density was adjusted to the desired concentration and seeded into 6-well or 24-well culture plates. After 6 h, the medium was replaced. Once the cell confluency reached 60–70%, the medium was refreshed again, and transfection was carried out 2 h later utilizing Lipofectamine^®^ 3000 Reagent (Invitrogen, San Diego, CA, USA). Each group was set up in triplicate for technical replicates. Twenty-four hours post-transfection, dual-luciferase activity was assessed utilizing the Dual-Luciferase^®^ Reporter Assay System (10-Pack Kit, Promega, Madison, WI, USA).

### 3.3. Recombinant Plasmid Construction

The coding sequences (CDS) of the *MK3* gene (Accession No. NM_001321558.2) and the *WT1* gene (Accession No. XM_040700507.2) were obtained from the NCBI database. Primers for amplifying the core coding regions of *MK3* and *WT1* were designed using the NCBI Primer-BLAST tool (see Appendix A). PCR products were purified and digested with *KpnI* and *XhoI* (NEB, Ipswich, MA, USA), then ligated into the pcDNA3.1(+) expression vector. To assess the regulatory elements of the *MK3* promoter, six deletion constructs were generated by designing primers to amplify promoter fragments located within the −2009 to +183 bp region upstream of the *MK3* CDS. The resulting PCR products were purified, digested with *KpnI* and *XhoI* (NEB, MA, USA), and ligated into the pGL3-Basic luciferase reporter vector. Primer sequences are provided in Appendix A.

### 3.4. RNA Interference

siRNAs targeting the *MK3* gene were designed and synthesized by Suzhou Jima Gene Co., Ltd. (Suzhou, China). The sequences of the siRNAs are listed in Appendix A. Transfection of granulosa cells with the siRNAs was performed utilizing the siRNA-Mate Plus transfection reagent.

### 3.5. Quantitative Real-Time PCR

Granulosa cells isolated from ovarian follicles of laying hens were seeded into 6-well plates. Once cells reached 60–70% confluence, transfection was performed. Twenty-four hours post-transfection, total RNA was extracted using TRIzol reagent (Invitrogen, CA, USA), and RNA integrity was confirmed by agarose gel electrophoresis. Reverse transcription was carried out utilizing the HisyGo RT Red SuperMix kit (Vazyme Biotech, Nanjing, China). qPCR was performed with the Taq Pro Universal SYBR qPCR Master Mix (Vazyme Biotech, Nanjing, China), using six technical replicates for each target gene. Primer sequences used for qPCR are listed in Appendix A. Gene expression levels were normalized to *GAPDH* expression and calculated using the 2^−ΔΔCt^ method [11].

### 3.6. Western Blotting Analysis

Granulosa cells isolated from the ovarian follicles of laying hens were seeded into 6-well plates. After 48 h, total cellular proteins were extracted utilizing RIPA lysis buffer supplemented with PMSF (Beyotime, Beijing, China). Protein concentrations were quantified using the BCA Protein Assay Kit (Vazyme Biotech, Nanjing, China). Proteins were separated via SDS-PAGE and transferred onto PVDF membranes. Before detecting the primary antibody, we first sectioned the Western blotting membrane of the target protein band. Membranes were blocked with a rapid blocking solution (Biosharp, Hefei, China) and then incubated overnight at 4 °C with the following primary antibodies: ASK1 (1:500, Proteintech Group, Cook County, IL, USA), TNF-R1 (1:500, Proteintech Group, IL, USA), P38 (1:1000, Abclonal, Wuhan, China), JNK (1:1000, Abclonal, Wuhan, China), TRAF2 (1:10,000, Proteintech Group, IL, USA), FAS (1:500, Abclonal, Wuhan, China), MK3 (1:1000, Abclonal, Wuhan, China), C-MYC (1:1000, Abclonal, Wuhan, China), BCL2 (1:500, Abclonal, Wuhan, China), BAX (1:1000, Abclonal, Wuhan, China), and Caspase3 (1:500, Abclonal, Wuhan, China). After washing, membranes were incubated with the appropriate HRP-conjugated secondary antibody, including GAPDH (1:50,000, Abclonal, Wuhan, China) as the internal control, for 1 h at room temperature. Protein bands were visualized and quantified utilizing ImageJ 1.42q software (National Institutes of Health, Rockville, MD, USA), with expression levels normalized to GAPDH.

### 3.7. Cell Counting Kit-8 (CCK-8) Assay

Granulosa cells were seeded into 96-well plates and transfected once cell confluency reached 60–70%. At 12, 24, 48, 72, and 96 h post-transfection, 10 µL of CCK-8 reagent (Vazyme Biotech, Nanjing, China) was added to each well and incubated for 1 h. Absorbance was measured at 450 nm using a PerkinElmer 2030 Multilabel Reader (PerkinElmer, San Diego, CA, USA) to evaluate cell viability.

### 3.8. 5-Ethynyl-2′-Deoxyuridine (EdU) Assay

Granulosa cells were plated in 6-well plates and transfected when 60–70% confluency was achieved. After 24 h, cells were incubated with 20 µM EdU (Beyotime, Shanghai, China) for 2 h. Following incubation, cells were fixed with fixation solution (Beyotime, Shanghai, China) for 15 min and permeabilized for another 15 min. The Click reaction solution was then applied and incubated in the dark for 30 min, followed by nuclear staining with Hoechst 33342 for 10 min. Images were captured using a fluorescence microscope (ECLIPSE Ti2, Nikon, Tokyo, Japan) from six randomly selected fields per sample. The proportion of EdU-positive cells (red fluorescence) relative to the total number of nuclei (blue fluorescence) was quantified using ImageJ software.

### 3.9. Apoptosis and Cell Cycle Analysis

Granulosa cells isolated from laying hen ovarian follicles were seeded into 6-well plates. When the cell density reached 60–70%, transfection was carried out. After 24 h post-transfection, apoptosis and cell cycle distribution were assessed utilizing the Annexin V-FITC Apoptosis Detection Kit (KeyGEN BioTECH, Nanjing, China). Flow cytometry was used for detection, and data were analyzed using Kaluza 2.1 flow cytometry software.

### 3.10. Data Statistics and Analysis

All experimental data were analyzed utilizing Microsoft Excel for one-way analysis of variance (ANOVA). For pairwise comparisons between groups, two-tailed *t*-tests were performed. Graphs were generated using GraphPad Prism 8.0 (GraphPad Software, San Diego, CA, USA). Results are presented as mean ± standard error of the mean (SEM). Statistical significance was set at * *p* < 0.05, while ** *p* < 0.01 was considered highly significant.

## 4. Results

### 4.1. Construction and Validation of an MK3 Gene Overexpression Vector

Based on prior RNA-seq analysis conducted by our research group, *MK3* was identified as a differentially expressed gene of interest. To explore its regulatory role in granulosa cells of laying hen ovarian follicles, we constructed an overexpression vector, pcDNA3.1-MK3 (Figure 1a,b). Granulosa cells were transfected with either the pcDNA3.1-MK3 plasmid (experimental group) or the empty pcDNA3.1 vector (control group). After 24 h, *MK3* expression levels were assessed. qPCR analysis revealed that *MK3* mRNA expression was significantly upregulated in the pcDNA3.1-MK3 group compared with the control group (*p* < 0.01; Figure 1c). Western blot analysis further confirmed the elevated protein expression of MK3 in the overexpression group, consistent with the qPCR results (Figure 1c,d). These findings indicate that the *MK3* overexpression vector was successfully constructed and effectively expressed in chicken granulosa cells.

### 4.2. Effect of MK3 Gene Overexpression on Granulosa Cell Proliferation and Apoptosis in Laying Hen Oocytes

To examine the regulatory effects of *MK3* on granulosa cell proliferation and apoptosis, granulosa cells were transfected with either the pcDNA3.1-MK3 plasmid or the empty pcDNA3.1 vector (control). After 24 h, gene and protein expression levels related to cell proliferation and apoptosis were evaluated. qPCR and Western blotting analyses revealed that overexpression of *MK3* significantly upregulated the pro-apoptotic genes *FAS* and *Caspase3* (*p* < 0.01), while downregulating the anti-apoptotic gene *BCL2* (*p* < 0.05) and the proliferation-related gene *C-myc* (*p* < 0.01) in granulosa cells (Figure 2a,c). To further validate these results, four siRNAs targeting *MK3* were designed, along with a negative control siRNA (NC-siRNA). Among them, MK3-siRNA-1192 showed the highest silencing efficiency (73%) and was selected for subsequent experiments. qPCR analysis confirmed that *MK3* knockdown reversed or attenuated the gene expression trends observed with overexpression (Figure 2b). Flow cytometry analysis of cell cycle progression (Figure 2d) showed that *MK3* overexpression significantly reduced the proportion of cells in the G1 phase (*p* < 0.01) and markedly increased the proportion in the S phase (*p* < 0.01), with almost no cells in the G2 phase. These results suggest that *MK3* arrests granulosa cell growth in the S phase and promotes apoptosis. Cell proliferation was further assessed utilizing the CCK-8 assay (Figure 2e). Compared to the control group, significant differences in cell viability were observed as early as 12 h post-transfection, with cell activity remaining significantly suppressed at 24, 48, 72, and 96 h (*p* < 0.01), indicating sustained inhibition of granulosa cell proliferation by *MK3*. Similarly, EdU incorporation assays (Figure 2f) demonstrated that *MK3* overexpression significantly reduced EdU-positive cell counts relative to the control group (*p* < 0.01), further confirming its inhibitory effect on granulosa cell proliferation.

To explore the pro-apoptotic role of *MK3*, apoptosis was measured using the Annexin V-FITC/PI double-staining method followed by flow cytometry. The results (Figure 2g) showed that *MK3* overexpression significantly decreased the overall cell survival rate and markedly increased early apoptosis levels compared to the control group (*p* < 0.01). Collectively, these findings demonstrate that *MK3* plays a crucial role in regulating granulosa cell fate by suppressing proliferation and promoting apoptosis in the ovarian follicles of laying hens.

### 4.3. Construction of the Deleted Fragment of the MK3 Gene Promoter and Screening of Transcription Factors

To explore the transcriptional regulation of the *MK3* gene, the nucleotide sequence of the 5′UTR, spanning from −2062 to +183 bp upstream of the transcription start site, was analyzed using an online promoter prediction tool. The results identified a core promoter region between −1480 and −418 bp. Further database searches via EPD and UCSC confirmed the presence of a cis-regulatory element (TATA box) at approximately −710 bp. To validate these predictions, six *MK3* promoter deletion constructs (K1–K6) were generated (Supplemental Appendix A) and evaluated for promoter activity utilizing the dual-luciferase reporter assay. As shown in Figure 3a, all *MK3* promoter deletion constructs exhibited significantly higher luciferase activity than the empty pGL3.0 control vector (*p* < 0.05). Notably, the fragment PGL-MK3-K2 (−1616/+183 bp) exhibited the highest promoter activity. In contrast, PGL-MK3-K5 (−515/+183 bp) showed significantly reduced activity compared with PGL-MK3-K4 (−875/+183 bp), suggesting the presence of essential promoter elements within the −875 to −515 bp region.

To further investigate transcription factors regulating *MK3* expression, transcription factor binding sites within the core promoter region (−875 to −515 bp) were predicted using bioinformatics tools (Appendix A). Among the predicted candidates, WT1 was identified as a potential transcription factor, with a predicted binding motif sequence of GTCCACCCCCGCAG located in this region. To assess the functional role of WT1 in regulating *MK3* transcription, overexpression and mutant constructs of the WT1 transcription factor were co-transfected with the *MK3* promoter-luciferase constructs into chicken follicular granulosa cells (Supplemental Appendix A). Dual-luciferase assays conducted 24 h post-transfection revealed that co-transfection with the WT1 overexpression vector significantly enhanced luciferase activity compared with the control group (*p* < 0.01; Figure 3b). However, co-transfection with the WT1 mutant vector did not produce a significant change in luciferase activity. In conclusion, these findings indicate that WT1 functions as a transcriptional activator of *MK3* by directly binding to its core promoter region, thereby regulating *MK3* gene expression in chicken granulosa cells.

### 4.4. The MK3 Gene Regulates Granule Cell Apoptosis Through the TNF/P38 MAPK Pathway

To investigate the molecular mechanism by which *MK3* regulates granulosa cell apoptosis in laying hen follicles, we focused on the classic TNF/P38 MAPK signaling pathway. Quantitative PCR analysis revealed that overexpression of *MK3* significantly increased the mRNA levels of key pathway genes, including *TRAF2*, *TNF-R1*, *ASK1*, *P38*, *JNK1*, and *BAX* (*p* < 0.05 or *p* < 0.01), while significantly reducing the expression of the anti-apoptotic gene *BCL2* (*p* < 0.05) (Figure 4a). Western blot analysis confirmed these transcriptional trends at the protein level (Figure 4a,c), suggesting that *MK3* may promote granulosa cell apoptosis by modulating downstream effectors such as *BAX* and *BCL2* within the TNF/P38 MAPK pathway. To validate this mechanism, granulosa cells were transfected with *MK3*-specific siRNA. As shown in Figure 4b, the knockdown of *MK3* reversed the expression trends observed in the overexpression group, with significantly reduced expression of *TRAF2*, *TNF-R1*, *ASK1*, *P38*, *JNK1*, and *BAX*, while *BCL2* showed no significant change. These results further confirm that *MK3* positively regulates granulosa cell apoptosis via the TNF/P38 MAPK pathway.

To further elucidate this regulatory mechanism, the ASK1-specific inhibitor GS-444217 was used to block signal transduction in the TNF/P38 MAPK pathway. Six concentrations of GS-444217 (0, 10, 20, 30, 40, and 50 μmol/L) were tested across different incubation times (1–6 h) following *MK3* overexpression for 24 h. Q-PCR analysis of *ASK1* mRNA levels (Figure 4d) showed a dose- and time-dependent decrease. At 40 μmol/L and 5 h of incubation, *ASK1* expression was almost completely suppressed, establishing this as the optimal inhibitory condition. Under these conditions, Q-PCR results further showed that mRNA levels of *P38* and *JNK1* were significantly downregulated after GS-444217 treatment (*p* < 0.01) (Figure 4e). Consistently, Western blot results demonstrated a marked reduction in ASK1 protein levels and significant decreases in *P38* and *JNK1* protein expression in the inhibitor-treated group compared to controls (*p* < 0.01) (Figure 4f). In summary, these findings confirm that the *MK3* gene mediates apoptosis in granulosa cells of laying hens via activation of the TNF/P38 MAPK pathway.

## 5. Discussion

The economic success of poultry production relies heavily on the proper functioning of ovarian follicles, particularly during the egg-laying period in hens. Granulosa cells play a pivotal role in follicular growth and development, and their physiological state directly influences reproductive efficiency. Granulosa cell fate is regulated by a complex interplay of extracellular apoptotic signals and intracellular pathways, including hormones, cytokines, oxidative stress, DNA damage, and apoptotic signaling cascades. These factors primarily modulate cell fate through the regulation of apoptosis-related proteins [12], making the molecular regulators of granulosa cell apoptosis promising targets for improving reproductive performance in poultry. *MK3*, a member of the MAPK family, is a serine/threonine kinase that translates extracellular stimuli into diverse cellular responses [13]. Previous studies in livestock have primarily focused on the role of *MK3* in follicular atresia, follicle growth and development, corpus luteum formation and regression, and oocyte maturation [10]. In this study, overexpression of the *MK3* gene in chicken granulosa cells resulted in significant upregulation of pro-apoptotic genes *FAS* and *Caspase3*, and downregulation of anti-apoptotic and proliferation-related genes *BCL2* and *C-myc*, respectively, as determined by qPCR. These results suggest that *MK3* promotes apoptosis granulosa cell. Functional assays, including CCK-8, EdU incorporation, and flow cytometry, further confirmed that *MK3* overexpression markedly inhibited granulosa cell proliferation. Moreover, cell cycle analysis indicated that *MK3* arrested the cell cycle in the S phase, thereby facilitating apoptosis in follicular granulosa cells. All these results indicated that *MKs* promotes apoptosis granulosa cell in chicken.

Gene expression in living organisms is regulated at multiple levels, with transcriptional regulation playing a central role in coordinating cellular responses to internal and external stimuli. To further study the regulation of upstream transcription factors on *MK3* genes, in this study, the upstream 5′UTR of the *MK3* gene was analyzed using an online promoter prediction tool. The analysis identified the core promoter region between −1480 and −418 bp and predicted a cis-acting element (TATA box) located at approximately −710 bp. Based on these predictions, we further verified that there was a functional promoter element within the −875 to −515 bp region and *WT1* was a key transcription factor. WT1 encodes a zinc finger protein that act as a multifunctional transcriptional regulator involved in cell proliferation, apoptosis, intercellular signaling, and cell adhesion. Previous studies have demonstrated that WT1 modulates the expression of genes associated with TGF-β, WNT, and MAPK signaling pathways, as well as genes involved in intercellular communication and ovarian steroidogenesis [14]. Additionally, WT1 is also been reported to regulate podocyte apoptosis and modulate p53 pathway-associated proteins in cells overexpressing BASP1 [15]; downregulation of WT1 induces apoptosis [16]. Recent studies further suggest that WT1 can be selectively degraded in granulosa cells through an autophagy-dependent mechanism [17,18]. Therefore, to validate the regulatory relationship between WT1 and *MK3*, both WT1 overexpression and mutant constructs were generated. Co-transfection experiments confirmed that WT1 specifically binds to the core promoter fragment PGL-MK3-K4, significantly enhancing *MK3* transcriptional activity. In conclusion, we successfully identified the core promoter region of the *MK3* gene and experimentally validated WT1 as a key transcription factor regulating its expression. These findings provide new mechanistic insights into the regulation of MK3 in granulosa cells.

Apoptosis is a programmed, non-inflammatory form of cell death and one of the most widely studied processes in cell biology. It is mainly initiated through two distinct pathways: the intrinsic (mitochondrial) pathway and the extrinsic (receptor-mediated) pathway. The intrinsic pathway is characterized by the release of cytochrome c from mitochondria into the cytosol, which subsequently activates a cascade of caspases—most notably Caspase-3—leading to cellular apoptosis [19]. In contrast, the extrinsic pathway is triggered by the activation of death receptors, such as TNFRs, located on the cell surface. Upon stimulation by external factors such as cytokines, these receptors initiate caspase activation and induce apoptosis [20]. A hallmark of apoptosis is the tightly regulated activity of caspases and changed the expression of pro-apoptotic and anti-apoptotic proteins [21]. Among the many apoptotic signaling cascades, the TNF/P38 MAPK pathway is particularly significant. In this pathway, TNF-α binds to TNFR1, initiating a signaling cascade that regulates cell fate. TNF-α is highly expressed in ovarian tissue and is distributed across multiple follicular compartments. Notably, its expression levels are significantly elevated in atretic follicles compared to normally developing ones [22], implicating TNF-α as a potential promoter of follicular atresia. Further evidence from porcine models demonstrates that during follicular atresia, the mRNA expression levels of *TNF-α* and *TRAF2* are markedly increased in the outer layer of granulosa cells [23]. Additionally, BCL-2 protein family play an important role In TRAIL-induced apoptosis involves intracellular apoptotic regulators [24].

This study demonstrated that overexpression of the *MK3* gene significantly increased the expression of key components of the TNF/P38 MAPK pathway, including *TNF-R1*, *TRAF2*, *ASK1*, *P38*, *JNK1*, and *BAX*, while decreasing the expression of the anti-apoptotic gene *BCL2*. In contrast, *MK3* knockdown via siRNA-1192 reversed the expression patterns of these genes, except for *BCL2*, which remained unchanged. These results suggest that *MK3* promotes granulosa cell apoptosis by modulating the expression of pro-apoptotic and anti-apoptotic genes—particularly *BAX* and *BCL2*—through the TNF/P38 MAPK pathway. This mechanism was further confirmed through pharmacological inhibition using GS-444217, a selective inhibitor of *ASK1*, a central regulator in the TNF/P38 MAPK cascade. *BCL2* is a well-established anti-apoptotic gene that support cell proliferation and follicular development [25]. Its role in follicular dynamics is well documented: *BCL2* deficiency leads to a significant reduction in the number of primordial oocytes and follicles in mice, alongside increased *BAX* expression [26]. Furthermore, *BAX* expression is significantly elevated in the granulosa cells of atretic follicles compared to healthy ones [27]. Similar patterns have been observed in porcine granulosa cells, where *BAX* levels were consistently higher in atretic follicles [28]. Collectively, our results findings demonstrate that *MK3* promotes granulosa cell apoptosis by regulating the balance between *BAX* and *BCL2* expression through the TNF/P38 MAPK pathway.

## 6. Conclusions

This study elucidates the molecular mechanism of how the *MK3* gene promotes apoptosis and suppresses cell cycle progression in granulosa cells of laying hen follicles. We also demonstrated that the transcription factor *WT1* directly binds to the *MK3* promoter and enhances its transcriptional activity. Furthermore, *MK3* activates the TNF/P38 MAPK pathway, leading to upregulation of downstream effectors such as *TRAF2* and *BAX*, which drives granulosa cell apoptosis in chicken (Figure 5). These findings will help us to better understand the role of *MK3* in follicle development and provides novel insights to explore the molecular mechanism of follicle development and production performance in chickens.

## Figures and Tables

**Figure 1 cells-14-01630-f001:**
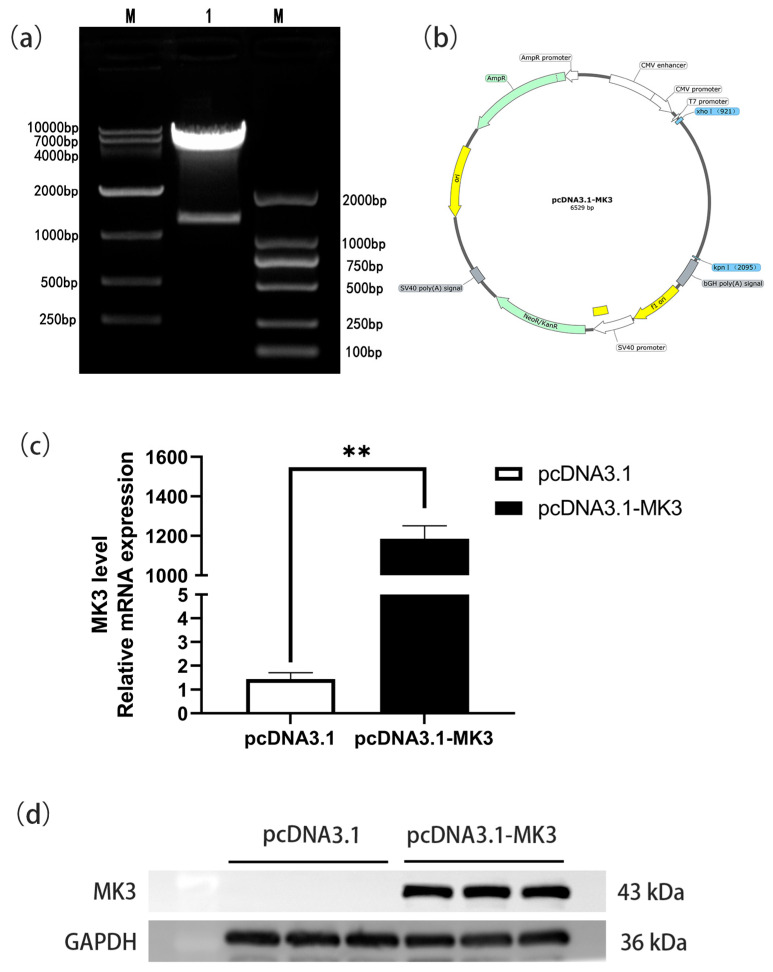
Construction and validation of the MK3 gene overexpression vector for chicken. (**a**,**b**) The double restriction enzyme digestion pattern and plasmid map of the pcDNA3.1-MK3 overexpression vector. (**c**,**d**) The mRNA (**c**) and protein expression (**d**) levels of MK3 in granulosa cells following transfection pcDNA3.1-MK3; ** Indicates extremely significant difference.

**Figure 2 cells-14-01630-f002:**
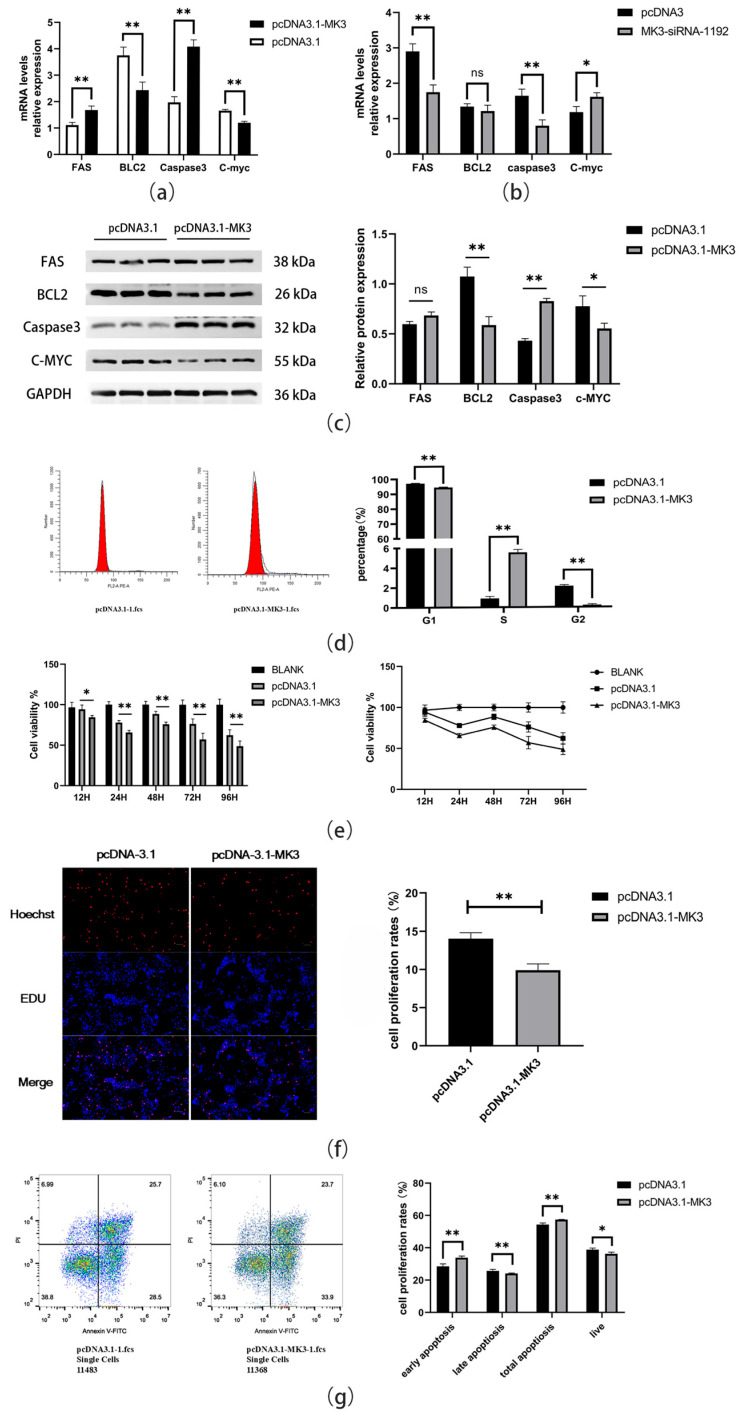
*MK3* gene upregulated the apoptosis in laying hen granulosa cells. (**a**,**b**) The mRNA expression levels of proliferation- and apoptosis-related genes after *MK3* overexpression (**a**) and *MK3* interference (**b**). (**c**) The FAS, BCL2, Caspase3, and C-MYC protein expression levels were detected by Western blot in chicken granulosa cells. (**d**) The cell cycle arrest profiles upon *MK3* overexpression. (**e**) Cell proliferation analyzed by the CCK-8 assay. (**f**) The apoptosis results of *MK3* overexpression in GCs by EdU assay. (**g**) Apoptosis in each group detected by annexin V-FITC/PI double staining; * Indicates significant difference, ** indicates extremely significant difference.

**Figure 3 cells-14-01630-f003:**
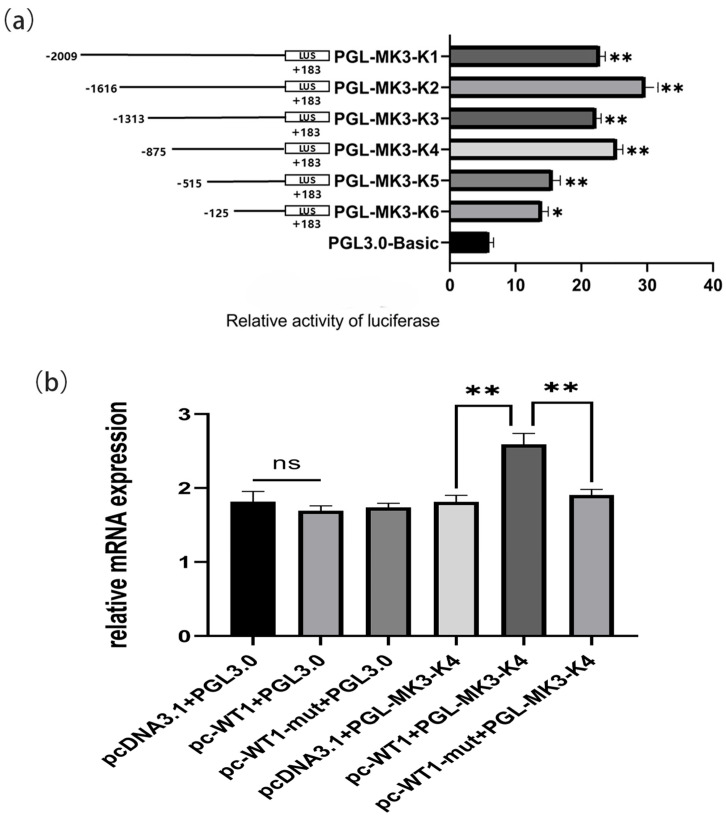
Identification of WT1-binding site in MK3 promoter region. (**a**) Luciferase assays show the activity of a series of deletion constructs in chicken GCs; (**b**) the dual-luciferase activity assay results were confirmed the binding interaction between WT1 and the MK3 core promoter fragmentt; * Indicates significant difference, ** indicates extremely significant difference.

**Figure 4 cells-14-01630-f004:**
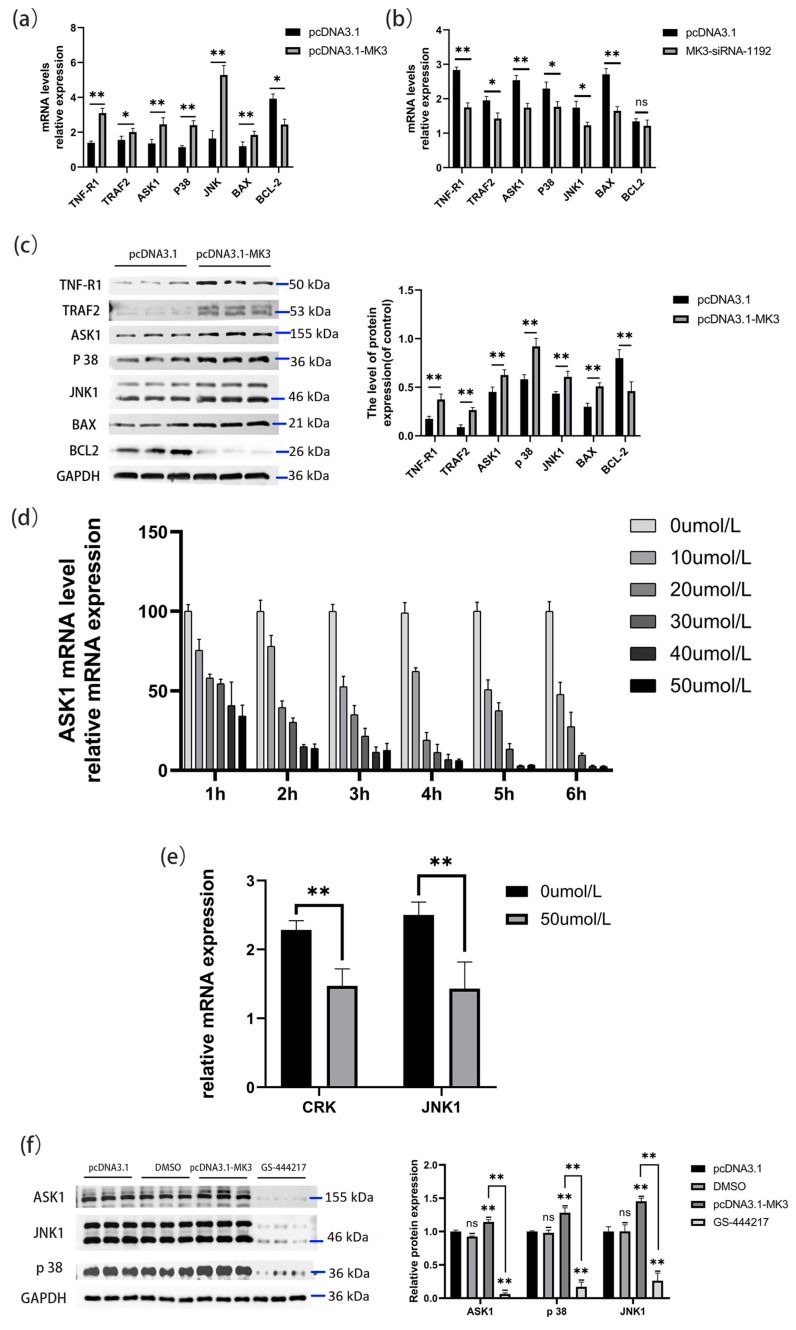
MK3 regulates granulosa cell apoptosis by TNF/P38 MAPK pathway. (**a**,**b**) The key genes (TNF-R1, TRAF2, ASK1, p38, JNK, BAX, and BCL2) mRNA expression levels of TNF/P38 MAPK pathway were detected by Q-PCR following MK3 overexpression (**a**) and siRNA interference (**b**). (**c**) TNF-R1, TRAF2, ASK1, p38, JNK, BAX, and BCL2 protein expression levels following MK3 overexpression were detected by Western blot. (**d**) The ASK1 mRNA expression levels which treated with different concentrations of GS-444217 and incubated in different times were detected by Q-PCR. (**e**) The mRNA expression levels of downstream genes P38 and JNK1 after inhibitor treatment were detected by Q-PCR. (**f**) The protein expression levels of ASK1, JNK1, and p38 were detected after GS-444217 treatment by Western blot, which were components of the TNF/P38 MAPK pathway; * Indicates significant difference, ** indicates extremely significant difference.

**Figure 5 cells-14-01630-f005:**
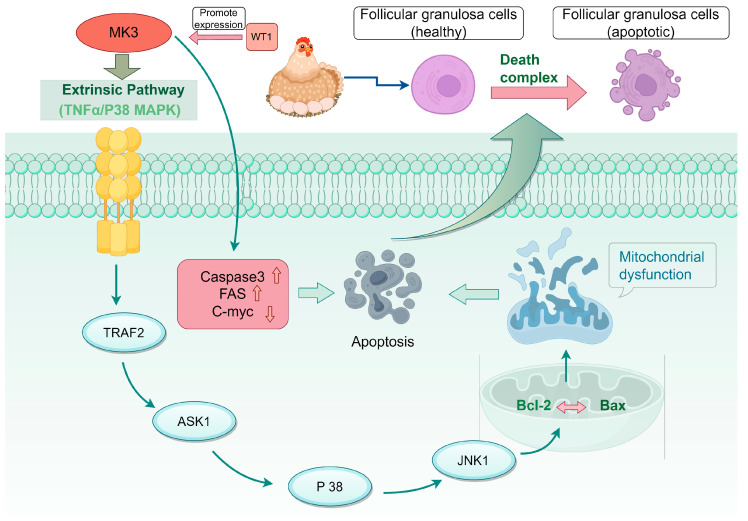
Schematic diagram of MK3 regulatory mechanism in chicken granulosa cells and follicle development.

## Data Availability

No data are stored in an official repository. Data/models supporting the research results are available to reviewers or can be obtained from the authors upon request.

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
