# Peer review of "MK3* Gene Upregulates Granulosa Cell Apoptosis Through the TNF/P38 MAPK Pathway in Chicken"

_cells, 2025, doi:10.3390/cells14201630_

Round 1

Reviewer 1 Report

Comments and Suggestions for Authors

This manuscript represents a well-designed and executed study on the role of MK3 in regulating granulosa cell apoptosis in laying hens, using a comprehensive approach including overexpression, siRNA knockdown, promoter mapping, and pathway inhibition. The work is novel in forwarding MK3 research from mammals to poultry and provides valuable information and suggests its potential role in reproductive efficiency. Overall, i have a couple of comments primary related to the interpretation issues rather than discrepancies in experimental design:

  • The study uses granulosa cells from 28-week hens from 1 source. The authors should discuss whether findings are generalizable across different breeds, ages, or physiological states.
  • The collected data are based on in vitro granulosa cell cultures. In vivo validation in ovarian tissue couldstrengthen the conclusions. Alternatively, the limitation should be acknowledged.
  • The use of ANOVA and t-tests is appropriate but does not address multiple comparisons. Did the authors use any correction (such as Bonferroni, Tukey)?
  • Are breeding implications premature at this point?

Author Response

Thank you very much for taking the time to review this manuscript. We extensively revised the manuscript according to the comments,and the detailed corrections are listed below point by point. Please find the detailed responses below and the corresponding revisions in the re-submitted files.

Comments 1: The study uses granulosa cells from 28-week hens from 1 source. The authors should discuss whether findings are generalizable across different breeds, ages, or physiological states.

Response 1: The granulosa cells which we used in this paper were randomly from different breeds, ages or physiological states. Thank you for pointing this out. We agree with this comment. Therefore, we have changed “Tissues were collected from healthy laying hens of similar age (28 weeks) and comparable egg-laying performance.” to “Follicles of various grades were collected from healthy laying hens.” in line 42 of page 2.

Comments 2: The collected data are based on in vitro granulosa cell cultures. In vivo validation in ovarian tissue could strengthen the conclusions. Alternatively, the limitation should be acknowledged.

Response 2: In this paper, we have elucidates the molecular mechanism by the MK3 gene promotes apoptosis and suppresses cell cycle progression in granulosa cells of laying hen follicles. In vivo validation in ovarian tissue will be carried out in the follow-up study.

Comments 3: The use of ANOVA and t-tests is appropriate but does not address multiple comparisons. Did the authors use any correction (such as Bonferroni, Tukey)?.

 Response 3: Yes, the Duncan correction were used in this paper.

Comments 4: Are breeding implications premature at this point?

 Response 4: Yes, the results of this study are provide a foundational framework for understanding the molecular mechanisms underlying follicle development in chicken. Therefore, we have changed the sentence in line 35-37 of page 1 to “Our findings highlight the key role of MK3 in follicular atresia, providing a molecular theoretical basis for improving reproductive performance and developing efficient molecular breeding strategies.”

Reviewer 2 Report

Comments and Suggestions for Authors

The manuscript presents interesting problem, however needs important changes.

  1. the title "Study..." needs to be changed, scientific manuscript should has title pointed out important, novel findings, not just study;
  2. the M&M section needs more details: a/strain of the hens; b/how many birds were in each group (not replication);c/description of the methods;d/ take out sentence like: "When the cells reached 60–70% confluency, transfection was performed" -it should be written only once.
  3. presentation of the results: a/ description on the figures must be corrected; b/ some of the figs. are almost not visible - 4d.
  4. discussion-it is repetition of the results but not discussion -trying to answer to questions: why? what was the reason of changes? and the most important-how to use this result in future, what next?
  5. Conclusion -it is again repetition of the results. The sentences:"These findings reveal the pivotal role of MK3 in follicular atresia and offer valuable molecular targets for improving reproductive traits in poultry. The work provides a theoretical foundation for the development of molecular breeding strat-egies aimed at enhancing egg production performance and boosting the economic effi-ciency of the poultry industry." are important but there is lack of the answer to question: how the improvement will be possible, what will Authors suggest to do next.

In summary: manuscript is interesting, Authors used many novel technics, methods, results are promising  but it must be corrected.

Author Response

Thank you very much for taking the time to review this manuscript. We extensively revised the manuscript according to the comments,and the detailed corrections are listed below point by point. Please find the detailed responses below and the corresponding revisions in the re-submitted files.

Comments 1: the title "Study..." needs to be changed, scientific manuscript should has title pointed out important, novel findings, not just study.

Response 1: Thank you for pointing this out. We agree with this comment. Therefore, we have changed the title of this paper to “MK3 gene upregulates chicken granulosa cell apoptosis by the TNF/P38 MAPK pathway.”

Comments 2: the M&M section needs more details: a/strain of the hens; b/how many birds were in each group (not replication); c/description of the methods; d/ take out sentence like: "When the cells reached 60–70% confluency, transfection was performed" -it should be written only once.

 Response 2: a: we added the information of strain of the hens in line 42 and line 46 in page 2;

b and c: in this study, the birds were only used to provide the follicle to isolate granulosa cells. So, 2-3 birds were selected at a time.

d: we have deleted the sentence "When the cells reached 60–70% confluency, transfection was performed" in line 41 for page 3.

Comments 3: presentation of the results: a/ description on the figures must be corrected; b/ some of the figs. are almost not visible - 4d.

 Response 3: we have corrected the description on the figure 1-4; b: we have adjusted and replaced Fig 4d.

Comments 4: discussion-it is repetition of the results but not discussion -trying to answer to questions: why? what was the reason of changes? and the most important-how to use this result in future, what next?

 Response 4: we have revised the discussion from line 11 for page 12 to line 12 for page 13. Next, for the application of the results of this study, we will further verify the role of MK3 in vivo.

Comments 5: Conclusion -it is again repetition of the results. The sentences:"These findings reveal the pivotal role of MK3 in follicular atresia and offer valuable molecular targets for improving reproductive traits in poultry. The work provides a theoretical foundation for the development of molecular breeding strat-egies aimed at enhancing egg production performance and boosting the economic efficiency of the poultry industry." are important but there is lack of the answer to question: how the improvement will be possible, what will Authors suggest to do next.

 Response 5: we have revised the conclusion in line 44 for page 13 to line 3 for page 14. In addition, we have added a graphic abstract in this part as figure 5.

Reviewer 3 Report

Comments and Suggestions for Authors

The introductory text provides the necessary background information required to understand the study. The references cited are appropriate. The study aim(s) is not clearly stated.

The methods and materials are well described and would enable replication of the study.

The results are clearly presented and the descriptions in the text reflect what is show in the figures.

The discussion is relevant and places the results in the context of the existing literature.

The conclusions are supported by the presented data.

Comments and suggestions:

Page 1

Abstract

I would suggest the authors add a sentence after the first sentence that describes problem/question this study aims to address.

Page 2

Paragraph 2

Line 6 If the authors previous RNAseq analysis has been published, it should be cited here.

Line 8 Please provide appropriate references for the studies in other livestock.

Paragraph 3

I would like to suggest that the authors revise this paragraph to state the aims of their study, rather than summarising it.

Page 3

Paragraph 2 The authors refer to Supplementary Table S1 – there are no supplementary tables provided with the manuscript. Two supplemental files are linked to the manuscript labelled “cells-3863550-supplementary” and “cells-3863550-original-images”. However, these contain the same folder labelled “Figure” which contains the composite images use for the figures in the manuscript.

There are four supplementary tables cited in the manuscript that were not available for review.

None of the uncropped images used in the study were provided for review.

Paragraph 4

Line 9 Please provide an appropriate citation for the delta-delta Ct method used in this study.

Page 5

Figure 1 The letter labels for each panel should be to the top left of each image. Same comment for all figures.

Fig 1d – The background of the two Western blots appears to be quite different. While the authors suggest that they have provided the original images as supplemental files, they have provided the high-resolution files of the complete figures, as shown here. The expectation is that the original files that have not been edited/manipulated in anyway should be provided as supplemental files.

Did the authors section the Western blotting membranes prior to probing with the primary antibodies or are these images cropped from complete membranes?

If the membranes were cropped prior to probing, I would suggest this be stated in the materials and methods.

There is the suggestion of bands being cropped off the GAPDH membrane – please comment.

Page 7

Figure 2 – See comments for Figure 1 regarding labelling.

Fig. 2c – There is the suggestion of bands being cropped off the BCL2 and Capase3 Western membranes – please comment.

Fig. 2e – What is the justification for showing this data in two formats?

Page 9

Figure 3

Fig 3a and 3b – These panels could be provided as supplemental data.

Fig 3d and 3e – These panels could be provided as supplemental data.

Page 11

Figure 4 – similar comments to other figures around labelling.

Fig 4c and 4f Several targets in these panels have multiple reactive bands, were all reactive bands used to develop the quantitative analyses shown in the associated graphs?

Fig 4f – As p38 and GAPDH are of similar sizes, how did the authors perform these Westerns?

Duplicate membranes?

Striping and re-probing?

Whatever approach was used, it should be detailed.

Author Response

Thank you very much for taking the time to review this manuscript. We extensively revised the manuscript according to the comments,and the detailed corrections are listed below point by point. Please find the detailed responses below and the corresponding revisions in the re-submitted files.

Comments 1: Page 1, Abstract, I would suggest the authors add a sentence after the first sentence that describes problem/question this study aims to address.

Response 1: Thank you for pointing this out. we have added “MAPK-activated protein kinase 3 (MK3) is a member of the mitogen-activated protein ki-nase (MAPK) family, which plays an important role in follicular development. Our previous RNA-seq analysis revealed that MK3 expression was significantly altered in the ovaries of laying hens exposed to normal versus light-deprivation conditions.” in the abstract.

Comments 2: Page 2, Paragraph 2, Line 6: If the authors previous RNAseq analysis has been published, it should be cited here. Line 8: Please provide appropriate references for the studies in other livestock.

Paragraph 3, I would like to suggest that the authors revise this paragraph to state the aims of their study, rather than summarising it.

 Response 2: Page 2, Paragraph 2, Line 6: The RNA-seq results and analysis are not published.

Line 8: we have added the references of other livestock in line 24 for page 2.

Paragraph 3: we have revised this paragraph. Listed as “In this study, based on the RNA-seq results of ovaries for normal- and light-deprivation conditions, we selected the different expression gene MK3 as the target gene to clarify its role in follicular development of chicken. Our work aims to confirm the molecular mechanism by which MK3 modulate follicular development when light changes and provide a novel insight into the regulation of production performance in chickens.” in the paper.

Comments 3: Page 3, Paragraph 2: The authors refer to Supplementary Table S1 – there are no supplementary tables provided with the manuscript. Two supplemental files are linked to the manuscript labelled “cells-3863550-supplementary” and “cells-3863550-original-images”. However, these contain the same folder labelled “Figure” which contains the composite images use for the figures in the manuscript.

There are four supplementary tables cited in the manuscript that were not available for review.

None of the uncropped images used in the study were provided for review.

Paragraph 4, Line 9: Please provide an appropriate citation for the delta-delta Ct method used in this study.

 Response 3: Page 3, Paragraph 2: we have rename the uploaded attachment and upload Supplementary Table S1-Table S4 as Supplementary Files. We have also upload the uncropped images as Figures files. 

Paragraph 4, Line 9: we have added the reference for the delta-delta Ct method used in this study.

Comments 4: Page 5, Figure 1: The letter labels for each panel should be to the top left of each image. Same comment for all figures.

Fig 1d – The background of the two Western blots appears to be quite different. While the authors suggest that they have provided the original images as supplemental files, they have provided the high-resolution files of the complete figures, as shown here. The expectation is that the original files that have not been edited/manipulated in anyway should be provided as supplemental files.

Did the authors section the Western blotting membranes prior to probing with the primary antibodies or are these images cropped from complete membranes?

If the membranes were cropped prior to probing, I would suggest this be stated in the materials and methods.

There is the suggestion of bands being cropped off the GAPDH membrane – please comment.

 Response 4: Page 5, Figure 1: we have revised the letter labels to the top left of each image for Figure 1 to Figure 4.

Fig 1d: we have provided the original files that have not edited/manipulated as supplemental files.

Yes, we have section the Western blotting membranes prior to probing with the primary antibodies.

We have added “Before detecting the primary antibody, we first sectioning the Western blotting membrane of the target protein band.” in the materials and methods in line 44 to line 45 for page 3.

The original images of the GAPDH membrane in Figure 1d was attached mistake. The correct original image is listed below and we have corrected it in the uncropped images supplementary files. Below the target band of GAPDH, there are some impurity bands.

Comments 5: Page 7, Figure 2 – See comments for Figure 1 regarding labelling. Fig. 2c – There is the suggestion of bands being cropped off the BCL2 and Capase3 Western membranes – please comment. Fig. 2e – What is the justification for showing this data in two formats?

Response 5: Page 7, Figure 2 - we have revised the letter labels to the top left of each image for Figure 1 to Figure 4.  

Fig. 2c - The original images of the Capase3 membrane in Figure 2c was attached mistake. The correct original image is listed below and we have corrected it in the uncropped images supplementary files.in addition, above the target band of BCL2 and Capase3, there are some impurity bands.

Comments 6: Page 9, Figure 3: Fig 3a and 3b – These panels could be provided as supplemental data. Fig 3d and 3e – These panels could be provided as supplemental data.

Response 6: we have listed the Fig 3a, 3b, 3d and 3e in the supplemental Figure S1 .

Comments 7: Page 11, Figure 4 – similar comments to other figures around labelling.

Fig 4c and 4f Several targets in these panels have multiple reactive bands, were all reactive bands used to develop the quantitative analyses shown in the associated graphs?

Fig 4f – As p38 and GAPDH are of similar sizes, how did the authors perform these Westerns?

Duplicate membranes?

Striping and re-probing?

Whatever approach was used, it should be detailed.

Response 7: Figure 4 –we have revised the letter labels to the top left of each image for Figure 1 to Figure 4.

Fig 4c and 4f- we have random selected one of the multiple reactive bands used to develop the quantitative analyses shown in the associated graphs, because all the bands have the same expression trend.

Fig 4f –we have used duplicate membranes.

In order to ensure the accuracy of the experiment, we made two pieces of glue at the same time, and repeated sample verification was carried out on each piece of glue with the same batch of protein samples. The whole subsequent experimental process was carried out synchronously on one instrument. 

Round 2

Reviewer 2 Report

Comments and Suggestions for Authors

Dear Authors,

I do not have further comments, but please:

1.Fig.2 f- take out the Chinese description

2. Fig.2 g-take out Lorem ipsem

3. Conclusion (s)- should not start with: "In summary.." because they have different meanings.

Author Response

Thank you very much for taking the time to review this manuscript. We extensively revised the manuscript according to the comments,and the detailed corrections are listed below point by point. Please find the detailed responses below and the corresponding revisions in the re-submitted files.

Comments 1: Fig.2 f- take out the Chinese description

Response 1: We have deleted the Chinese description.

Comments 2: Fig.2 g-take out Lorem ipsem.

 Response 2: We have deleted the Lorem ipsem.

Comments 3: Conclusion (s)- should not start with: "In summary.." because they have different meanings.

Response 3: We have deleted “In summary…” on Coclusion(s).

Reviewer 3 Report

Comments and Suggestions for Authors

The authors have addressed the majority of the comments and suggestions I made on the submitted version of thier manuscript.

Note the original images for the Western blot images were not provided as stated in thier responses. The images provided are high-resolution composites used for each figure, identical to the original submissions.

Regarding my query about the quantification of bands on the Western blot membranes shown in Fig 4c and Fig 4f, where some targets yielded multiple bands, the authors responded that one reactive band was selected "at random", as all the reactive proteins show the same trends.

I tend to agree that this is the case for TRAF2 and JNK1 in Fig 4c. Similarly, for JNK1 in 4f. I am not convinced this argument holds for ASK1 in Fig 4f. I would strongly suggest that the authors clearly identify/mark the bands used for quantification in these images. I think this is essential for experimental repeatability and transparency. 

Author Response

Thank you very much for taking the time to review this manuscript. We extensively revised the manuscript according to the comments,and the detailed corrections are listed below point by point. Please find the detailed responses below and the corresponding revisions in the re-submitted files.

Comments 1: Note the original images for the Western blot images were not provided as stated in thier responses. The images provided are high-resolution composites used for each figure, identical to the original submissions.

Regarding my query about the quantification of bands on the Western blot membranes shown in Fig 4c and Fig 4f, where some targets yielded multiple bands, the authors responded that one reactive band was selected "at random", as all the reactive proteins show the same trends.

I tend to agree that this is the case for TRAF2 and JNK1 in Fig 4c. Similarly, for JNK1 in 4f. I am not convinced this argument holds for ASK1 in Fig 4f. I would strongly suggest that the authors clearly identify/mark the bands used for quantification in these images. I think this is essential for experimental repeatability and transparency.

Response 1: We have marked the bands for all proteins in Fig 4c and Fig 4f and have replaced Fig 4 in the paper.  

Round 3

Reviewer 3 Report

Comments and Suggestions for Authors

The authors have addressed the comments and suggestions I made on the revised version of thier manuscript.

I have no further comments.